# Effect of Alpha-Lipoic Acid Supplementation on Low-Grade Squamous Intraepithelial Lesions—Double-Blind, Randomized, Placebo-Controlled Trial

**DOI:** 10.3390/healthcare10122434

**Published:** 2022-12-02

**Authors:** Anja Divković, Kristina Radić, Damir Sabitović, Nikolina Golub, Marija Grdić Rajković, Ivana Rumora Samarin, Zinaida Karasalihović, Adnan Šerak, Emir Trnačević, Petra Turčić, Dražan Butorac, Dubravka Vitali Čepo

**Affiliations:** 1Department of Laboratory Diagnostics, University Clinical Centre Tuzla, 75000 Tuzla, Bosnia and Herzegovina; 2Faculty of Pharmacy and Biochemistry, University of Zagreb, 10000 Zagreb, Croatia; 3Faculty of Food Technology and Biotechnology, University of Zagreb, 10000 Zagreb, Croatia; 4Department of Obstetrics and Gynecology, Sestre Milosrdnice University Hospital Centre, 10000 Zagreb, Croatia

**Keywords:** low-grade squamous intraepithelial lesion, human papilloma virus, alpha lipoic acid, inflammation, dietary patterns

## Abstract

Low-grade squamous intraepithelial lesion (SIL) is a cytologic diagnosis etiologically related to human papilloma virus (HPV) infection that leads to the release of inflammation mediators, the formation of reactive oxygen species (ROS) and decreased levels of antioxidants in tissues, which is why antioxidants might be considered effective against SIL progression. This randomized double-blind placebo-controlled study aimed to investigate the effectiveness of alpha-lipoic acid (ALA) supplementation (600 mg/day) on the regression of low-grade SIL in 100 patients. Low-grade SIL was determined after the cytological screening, colposcopic examination and targeted biopsy and histological confirmation of cytological–colposcopic diagnosis. Inflammation parameters and the presence of HPV were determined by standard laboratory methods. Dietary and lifestyle habits were investigated using a standardized and validated semi-quantitative food questionnaire (FFQ). ALA supplementation significantly reduced the proportion of patients with low-grade cytological abnormalities, in comparison to placebo. Given the obtained level of significance (*p* < 0.001), the presented results indicate that short-term ALA supplementation shows a clinically significant effect on cervical cytology. Future studies should focus on the use of innovative formulations of ALA that might induce bioavailability and therapeutic efficiency against HPV infection and the investigation of synergistic effects of concurrent dietary/lifestyle modification and ALA supplementation in both low-grade and high-grade SIL.

## 1. Introduction

Low-grade squamous intraepithelial lesion (LSIL) is a cytologic diagnosis for patients with smears showing cytologic criteria of permissive HPV infection or cervical interepithelial neoplasia I (CIN I); the same classification is used for histopathologic diagnosis where low-grade SIL and low-grade CIN are used synonymously [1]. LSILs account for most of the cytological anomalies for screening cervical cancer and they usually regress spontaneously within a year from the diagnosis. However, the exact rates of spontaneous regression of LSIL are hard to predict due to the heterogenicity of conducted studies: regression percentages observed in different studies ranged between 7% and 95%. The rates of progression to a high-grade squamous intraepithelial lesion (HSIL) increase significantly if LSIL is caused by high-risk, oncogenic human papilloma virus (HPV) genotypes [2].

In the etiology of the formation of squamous intraepithelial lesions (SIL), HPV infection leads to the release of different inflammation mediators, resulting in the formation of reactive oxygen species (ROS) and causing a decrease in the level of antioxidants. HPV integration in combination with oxidative stress causes oxidative damage to the genome and different epigenetic alterations that altogether hinder apoptosis and alter cellular proliferation [3]. Therefore, diet components exerting antiviral, anti-inflammatory and antioxidant actions could be considered protective against the progression of SIL and the development of cervical cancer.

Observational studies confirm that Western dietary patterns increase the possibility of HPV infection, while high adherence to the Mediterranean diet decreases the risk [4]. Numerous observational studies indicate that the consumption of a wide variety of whole foods rich in vitamins, minerals and nonessential bioactive compounds (especially those with antioxidant and antiviral properties) can be effective in preventing the progression of LSILs to HSIL or cervical cancer, and, therefore, a balanced-diet prevention strategy should be recommended to patients upon LSIL diagnosis [3,5,6,7].

Investigating dietary supplementation with antioxidants (and other vitamins) for cervical cancer prevention and treatment has recently gained considerable interest [8]. Long-term oral beta carotene (30 mg, 2 years) or folate (10 mg, 6 months) did not significantly impact the regression of HSIL [9,10]. On the other hand, taking selenium supplements (200 μg for 6 months) among patients with LSIL led to its regression and had beneficial effects on their metabolic profiles [11]. Folate supplementation (5 mg/day, 6 months) was efficient in inducing the regression of LSIL [12], and vitamin D (50,000 IU every two weeks for 6 months) prevented the recurrence of HSIL [13].

Due to a low number of high-quality intervention studies, the level of evidence for the efficiency of vitamin/antioxidant supplements in LSIL is low and there are currently no official recommendations for patients. Therefore, additional investigations into efficient approaches for inducing the regression/preventing the progression of LSIL are necessary.

Alpha lipoic acid (ALA), also known as 1,2-dithiolane-3-pentanoic acid, thioctic acid, or its reduced form—dihydrolipoic acid, are potent nutritional antioxidants and perform pleiotropic actions on different pathways linked to numerous diseases, including direct antiradical activity or modulating the signalling transduction of several pathways (such as insulin and nuclear factor kappa B (NFkB)). It is, therefore, used as a potential therapeutic agent in diabetic neuropathy, brain disease and cognitive dysfunction, cardiovascular diseases, endothelial dysfunction, hemorrhoidal illness, obesity and cancer [14,15,16,17]. The majority of clinical studies conducted so far were focused on its antioxidant activity and potential effectiveness in diabetic neuropathy, Alzheimer’s disease and cancer [18].

As mentioned, the majority of its therapeutic effects are contributed to direct antioxidant activity—a scavenging capacity for ROS and metal chelating abilities. Additionally, ALA regulates numerous signalling pathways—insulin pathway, NFkB and adenosine monophosphatase protein kinase (AMPK)—but the clinical significance of these actions still needs to be confirmed [16].

This study aimed to investigate the impact of supplementing patients with diagnosed LSIL with alpha lipoic acid (ALA) and observe the effects on the progression/regression of LSIL after a 3-month supplementation.

## 2. Materials and Methods

### 2.1. Participants and the Study Design

Criteria for inclusion of patients in the study were histological confirmation of low-grade SIL and age (18–55 years). Exclusion criteria were pregnancy; malignant diseases; diabetes; chronic inflammatory diseases; hysterectomy, destructive therapy of the cervix, or previous abortion; HPV vaccination and menopause; and regular use of dietary supplements (except multivitamin/multimineral dietary supplements containing the amounts of nutrients below or equal to current recommended dietary allowances (RDAs)). Patients who met the above-mentioned criteria were introduced to the purpose and the manner of conducting the study and were included in the study after giving informed consent. Recruitment for the trial was performed at the University Clinical Centre Tuzla between January 2020 and March 2022.

The study was a double-blind (participants and care-providers), randomized, placebo-controlled trial, where 100 participants aged 19–51 years were asked to self-administer either 600 mg of alpha lipoic acid in the form of two 300 mg capsules (Zada Pharmaceuticals, Lukavac, Bosnia and Herzegovina) or a placebo (provided as identical oral capsules containing rice starch) per participant, per day for 3 months. Participants were distributed in a ratio of 1:1 for ALA:placebo. Block randomization was performed to ensure balance using an in-house computer program; patients were randomly allocated a patient number that determined their supplement allocation, and two bottles of capsules pre-labelled with their study number were dispensed. The two bottles contained sufficient capsules for 3 months. Compliance to treatment was encouraged by weekly phone calls and text messages. Patients were asked to return the remaining capsules after the 3-month period in order to assess the adherence. Active participation in the study ended on June 2022 after the last enrolled participant attended their 3-month appointment.

All participants gave their written informed consent for inclusion in the study. The informed consent and the research protocol were approved by the Ethics Committee of the University of Zagreb, Faculty of Pharmacy and Biochemistry (no: 251-62-03-18-23) and the Ethics Committee of the University Clinical Centre Tuzla (no: 02-09/2-61-16). An independent Data and Safety Monitoring Committee was in place throughout the trial to review the progress of the study and potential side effects. The trial is registered at ClinicalTrials.gov (Clinical-Trials.gov, number NCT05485259) and complies with the CONSORT guidelines. It was performed in accordance with the international, national and institutional guidelines pertaining to clinical studies and biodiversity rights.

### 2.2. Data Collection

All recruited patients had a confirmed diagnosis of LSIL. LSIL was determined after the performed cytological screening, colposcopic examination of the cervix and targeted biopsy and histological confirmation of cytological–colposcopic diagnosis. At the first appointment, patients were introduced to the study design, and they provided written consent to be enrolled in the study. They filled out a baseline questionnaire and provided information on reproductive history and current medication. Examination of patients’ dietary and lifestyle habits was conducted using a standardized and validated semi-quantitative food questionnaire (FFQ) [19] with the help of trained staff. The data were collected to obtain basic information on dietary patterns, antioxidant intake, use of dietary supplements, smoking, and physical activity of patients. Patients were advised not to change their diet and level of physical activity during the study. They received two bottles prelabelled with their study number, containing enough capsules of either ALA or placebo, sufficient for 3 months of therapy. The appearances of the bottles and capsules were identical in the two groups.

During the supplementation period, participants were contacted twice by telephone by a research team member to check possible adverse effects and to improve adherence. Three months after entry, participants were invited to a follow-up appointment, where cytological screening, colposcopic examination of the cervix and targeted biopsy and histological confirmation of cytological–colposcopic diagnosis were conducted. At both appointments (initial and 3 months appointment), blood samples were taken from the patients’ cubital vein by the standard. Blood samples were collected by venipuncture after a 10 h fasting period.

### 2.3. Assessment of Primary Outcomes

The primary outcome of the study was LSIL, which was determined after the performed cytological screening, colposcopic examination of the cervix and targeted biopsy and histological confirmation of cytological–colposcopic diagnosis at the study baseline and after the 3 months intervention. The findings of cytological screenings were qualified according to the internationally recognized form “Zagreb 2002” [20]. The colposcopic form “Rio de Janeiro-Zagreb 2011” (https://www.hdgo.hr/userFiles/upload/documents/aktualne-teme/kolposkopija/Rio-Zagreb-2011.pdf, accessed on 12 August 2021) was used to classify the colposcopic findings. Uterine tissue samples obtained by targeted colposcopic biopsy were processed by a standard histological method. Assessment of the pathological diagnosis was done as blindness by a single experienced pathologist at baseline and after the intervention. The classification of premalignant and invasive cervical squamous epithelial lesions was determined according to the WHO tumor classification [21].

### 2.4. Assessment of Secondary Outcomes

Secondary outcomes of the study were high-risk human papillomavirus (HPV) positivity after the 3-month follow-up visit and the biomarkers of inflammation (high-sensitive C-reactive protein, fibrinogen and sedimentation). Detection of the presence of high-risk HPV in cervical smears was made using the Hybrid Capture II HPV Test (Qiagen, Germantown, MD, USA). The samples were tested for the presence of HRHPV types 16, 18, 31, 33, 35, 39, 45, 51, 52, 56, 58, 59, and 68. For the assessment of other secondary outcomes blood samples were collected before and after the treatment at three months in the experimental group by venipuncture after a 10 h fasting period. Blood was placed in siliconized test tubes without anticoagulant for biochemical determinations. The blood sedimentation rate was measured according to standard protocol in BD Vacutainer^®^ blood collection tubes (Becton, Dickinson and Company, New York, USA). Fibrinogen was determined by standard methodology, using an automated hemostasis analyzer (BCS^®^ XP System, Siemens, Marburg, Germany). High-sensitive CRP was determined by the immunoturbidimetric method using Multigent CRP Vario^®^ reagent on Architect ci8200 integrated system (Abbott, Chicago, IL, USA)

### 2.5. Analysis of Diet Characteristics

A semiquantitative food frequency questionnaire (FFQ) was used for the assessment of food intake. FFQ was designed as a 192-item questionnaire with one month as the reference period of intake. There were 100 dietary items listed, and the remaining items were questions about supplementation use and eating habits. The questionnaire was constructed as a modification of a previously published questionnaire [22], with regard to serving sizes and the national specificity of foods. The participants filled out the questionnaire under the supervision of a researcher. Average daily food intake in grams, as well as in serving sizes, was calculated for each participant based on the frequency and portion size reported in the FFQ. Nutrient intake for selected nutrients was calculated using national food composition tables [23] and serving sizes, according to USDA Dietary Guidelines for Americans, 2020–2025 [24].

### 2.6. Sample Size Assessment and Statistical Analysis

To determine the sample size, we used a randomized clinical trial sample size formula where type one (α) error was set at 5% and the study power was set at 85%. The ratio of the case to control was 1, and the expected dropout was 2%. LSIL was the main outcome of the study; the expected proportion of LSIL regression in the control group was 50%, and in the treated group it was 80%. Assuming a dropout of 2% in the final sample size and superiority margin of 5%, the total population study was determined to be 96 (48 subjects per group) (http://www.riskcalc.org/samplesize/, accessed on 12 August 2021).

The main statistical analyses were performed on all randomized women for whom outcome data were available. The primary outcome was histologically confirmed LSIL vs. histology without LSIL, or negative colposcopy (and no histology). The secondary outcomes were HPV infection, sedimentation rate (SE) and fibrinogen (FI).

The comparison of binary outcomes between treatment groups was made by 2 × 2 tabulation and a risk ratio (RR), 95% confidence intervals (CIs) and calculated *p*-values. For the analysis of numerical variables, nonparametric statistical tests were used, due to a low number of patients in the placebo or the treated group (n ≤ 50), and the results were presented as medians and interquartile ranges. To identify statistically significant differences between the groups, the Mann–Whitney test for the independent group or the Wilcoxon’s test for the dependent group was used. The Fisher’s exact test was used for contingency analysis. Statistical analyses were done using GraphPad Prism 8.4.3 (GraphPad Software LLC, San Diego CA, USA) and MedCalc statistical software MedCalc ver.14.8.1.0., MedCalc Software Lcd., Ostend, Belgium). The *p* level <0.05 was considered statistically significant.

## 3. Results

A total of 100 patients were randomized into the placebo (n = 50) or the treatment group (n = 50). Eleven patients did not finish the trial due to personal reasons (two patients from the placebo group and nine patients from the treatment group). (Figure 1). No side effects were reported following the intake of ALA supplements or placebo throughout the study. Additional description of the study design and more detailed flow diagram of the progress through the phases of a trial (CONSORT checklist and CONSORT flow diagram) are presented as Appendix A, respectively (Appendix A). Raw experimental data obtained during the study are presented in Appendix A.

Numerous demographic, lifestyle and diet characteristics (such as age; body mass index (BMI); cigarette smoking; low intake of fruit and vegetables; and high intake of energy, red meat and animal protein) that have been determined as risk factors for the occurrence of gynecological abnormalities, including HSIL or cervical cancer [25,26,27], were compared between the two groups (Figure 2). The age of participants in the two study groups was similar (*p* = 0.0823). Medians of the body mass index (BMI) in the placebo and the intervention group were similar (23.89 and 24.98 kg/m^2^, respectively; *p* = 0.5086). The percentage of smokers in the placebo and the intervention group was 29.2% and 19.5%, respectively, and observed differences were not statistically significant (*p* = 0.3326). The intake of fruit and vegetables was significantly lower than recommended, and intake of meat and animal proteins was above current recommendations in both groups (US recommendations) and obtained medians of intake did not differ significantly between the placebo and the treated group of patients. Compliance was self-reported on the final visit, and, in addition, unused capsules were counted upon return. The medians of the numbers of returned capsules at the final visit were 35 and 28 in the placebo and the treated group, respectively, and observed differences were also not statistically different (*p* = 0.4054).

The primary and secondary outcomes of this study were diagnosis of LSIL (primary outcome), HPV infection, SE rate, hsCRP and FI (secondary outcomes), and they were compared between the placebo and the treated group at two time-points (initial visit and 3-month follow-up visit) (Table 1). Additionally, the differences between the values obtained in the placebo or the treated group at the initial and the 3-month follow-up visit were also compared and presented in Table 2. Odds ratio indicating the efficiency of 3 month ALA supplementation on the recovery of LSIL-positive patients and its precision are presented in Table 3.

All patients recruited for the study had a diagnosis of LSIL at the initial visit; by the end of the study, this number of patients was reduced in both groups—in the placebo group 44 patients still had an LSIL diagnosis; in the treated group of patients only 2 of them still had an LSIL diagnosis (*p* < 0.0001). The percentage of patients with positive HPV findings remained the same during the 3 months of study; 37.5% in the placebo and 46.3% in the treated group of patients were HPV-positive and observed differences were not statistically significant. At the initial visit, hsCRP and FI were significantly lower in the placebo group (*p* = 0.0344 and *p* = 0.0130, respectively) but still within the normal range, while SE rates were comparable (*p* = 0.0785). At the 3-month follow-up visit, the situation was reversed: inflammation parameters were higher in the placebo group and observed differences were statistically significant for FI and SE rates (*p* = 0.0437 and 0.0046, respectively).

Table 2 shows the significance of observed changes in primary and secondary outcomes that occurred during the 3 months of ALA/placebo supplementation. The number of LSIL-positive patients decreased from 48 to 44 in the placebo group (8.33% of patients recovered), but the observed change was found to be statistically insignificant (*p* = 0.1171). On the other hand, in the ALA-treated group as much as 95.12% of patients recovered (*p* < 0.0001). None of the patients in either of the investigated groups progressed to the higher grade of SIL. The percentage of HPV-infected patients remained the same in both groups. All inflammation parameters increased during 3 months of supplementation in the placebo group and observed changes were statistically significant (*p* < 0.001). On the contrary, in the ALA-treated group all observed inflammation parameters decreased moderately but significantly (*p* < 0.0001). There were no adverse effects reported in the study.

## 4. Discussion

As mentioned previously, investigations on the possibility of dietary intervention in LSIL are scarce and the obtained results are rather contradictory. This is the first investigation of the effectiveness of ALA in inducing the regression of LSIL. Results obtained within this research show that three-month supplementation with 600 mg of ALA significantly reduced the proportion of patients with low-grade cytological abnormalities, in comparison to placebo. Given the obtained level of significance (*p* < 0.001), the presented results indicate that short-term ALA supplementation might have a clinically significant effect on cervical cytology.

In the placebo group, spontaneous LSIL regression was observed in only 8.3% of patients, which is rather low considering that rates are predicted to be ranging from 40% to 70% over a period of 1–5 years [28]. Additionally, in available intervention studies that investigated the possibilities of nutritional supplementation in inducing the regression of LSIL, the rates of spontaneous regression observed in placebo groups were higher (52.0–56.0%) [11,12,13,28]. The reasons for the observed disagreement with the results obtained in this study might include the short monitoring period (3 months vs. 6 months in intervention trials or 1–5 years in observational studies) and a high median age of participants (37 and 41). Namely, age is considered an independent risk factor for LSIL progression, where with every five years of age, the odds for regression are reduced by 21% independently of CIN grade and presence of HPV high-risk infection [29].

The dietary habits of the participants included in this study might also have contributed to the low LSIL regression rates in the placebo group. Namely, the results of the recent review of Hui and co-authors [5] highlighted the benefits of high consumption of whole fruits and vegetables, nuts and fish in terms of their protectiveness against persistent HPV infection and SIL. On the other hand, the low consumption of whole vegetables and fruits was associated with a three-fold increase in the risk of CIN 2 and 3 in subjects with high HPV viral load. Therefore, patients with diagnosed SIL are always advised to increase their dietary intake of protective nutrients. Analysis of FFQs in this study showed a low intake of both fruits and vegetables that are known to be major protective factors against LSIL progression (2.78 and 3.49 portions per week, respectively). This intake remained low during the supplementation period as patients were advised not to change dietary habits during the period of intervention.

The response to supplementation in the treated group of patients was high; the regression of LSIL has been observed in 95% of participants, while the rates of HPV infection remained intact (19 patients that were HPV-positive at baseline remained positive despite supplementation). The obtained results are consistent with the results presented in several other clinical trials investigating the efficiency of nutrients in the induction of LSIL regression. Unlike ours, those studies included a lower number of patients and investigated long-term supplementation protocols (6 months). Vahedpoor and co-workers [13] showed that 6-month vitamin D supplementation of patients with LSIL (one dose of 50,000 IU vitamin D supplement every 2 weeks) resulted in the regression of cervical intraepithelial neoplasia grade 1 (CIN1) in 84.6% of patients (n = 29). Similarly, supplementation with 5 mg of folic acid for 6 months resulted in 83.3% regression of CIN 1 (n = 29) [12], as well as the long-term supplementation (6 months) with 200 µg of organic form of selenium in 88% of women in the supplemented group (n = 28) [11].

The mechanisms of potential efficiency of ALA in the regression of LSIL can only be speculated. Since ALA is a powerful antioxidant, the explanation of its protective mechanism can be similar to the protective effects of other dietary antioxidants against cervical cancer (inhibition of the proliferation of cancer cells, stabilization of the p53 protein, prevention of DNA damage and reduction in immunosuppression) [27]. The anti-inflammatory effects of ALA probably contribute to its efficiency, since inflammation is strongly associated with cancer and can predispose to tumours. Therefore, targeting inflammation and the molecules involved in the inflammatory process could represent a good strategy for cancer prevention and therapy [30]. A recent study showed that high hsCRP levels are significantly associated with a lower rate of SIL regression and suggested that monitoring hsCRP might help in monitoring LSIL [31]. ALA’s known anti-inflammatory efficiency has also been supported by the results of this study, as shown by a significant reduction in all investigated inflammatory parameters (hsCRP, FI, SE) in the supplemented group of patients (as opposed to the inflammatory markers in the placebo group). The obtained results are consistent with the latest literature data, even though available data regarding the effects of alpha-lipoic acid (ALA) supplementation on inflammatory markers are controversial, the recent meta-analysis showed the promising impact of ALA administration on decreasing inflammatory markers, such as CRP, IL-6 and TNF-α among patients with metabolic disorders [32]. Therapeutic regimes used in the studies included in the meta-analysis were comparable to ours (300–600 mg ALA, 2 weeks–12 months). This could also have contributed to its efficiency in inducing the regression of LSIL.

Observed protective effects of ALA showed in this study might also be explained by its antiproliferative actions investigated by other authors: the ability to induce apoptosis through activating pyruvate dehydrogenase in tumour cells, the ability to act on different signalling pathways (activation of AMPK and subsequent down-regulation of mTOR-S6 signalling pathway or Grb2-mediated EGFR down-regulation), or the induction of ROS production in tumour cells [33,34,35,36].

Even though ALA is known to have antiviral effects against some viruses [37], its efficiency is probably not due to antiviral properties. Namely, HPV rates remained intact in both investigated groups, which is consistent with the results of studying other antioxidants in the prevention of cervical cancer, obtained by other authors [38].

Despite obvious advantages (randomized, placebo-controlled design and relatively high numbers of participants, in comparison to other similar intervention studies), this investigation had some limitations. The major limitations are short-period supplementation (3 months, which was due to financial and organizational limitations) and a relatively high median age of patients recruited for the study, resulting in low rates of spontaneous regression in the placebo group. Additionally, due to budget limitations, it was impossible to monitor the serum ALA levels of the participants, which would contribute significantly to the quality of conducted research, especially if we consider the short half-life and bioavailability of ALA (about 30%) triggered by its hepatic degradation and reduced solubility, as well as instability in the stomach [18]. Future studies should focus on the use of innovative formulations of ALA that might induce the therapeutic efficiency of ALA against HPV infection and investigation of synergistic effects of dietary modification and ALA supplementation in both LSIL and HSIL regression.

## Figures and Tables

**Figure 1 healthcare-10-02434-f001:**
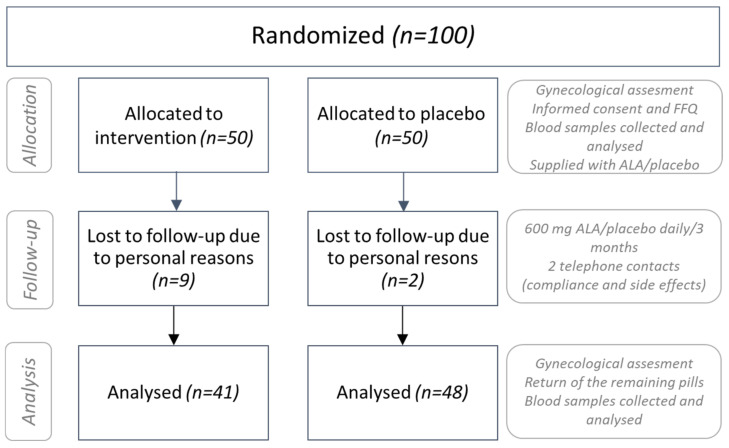
Patient flow diagram.

**Figure 2 healthcare-10-02434-f002:**
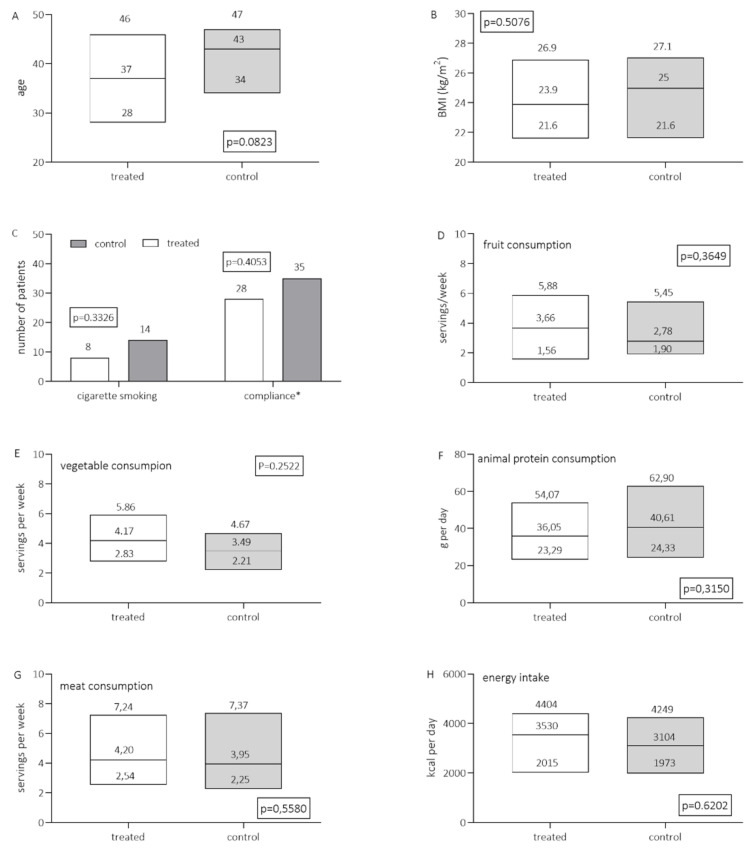
Characteristics of study groups at baseline regarding age (**A**), BMI (**B**), cigarette smoking and compliance (**C**) and dietary habits (**D**–**H**). Results are expressed as medians (min–max range) Results were tested by Mann–Whitney’s test (**A**, **B**, **D**–**H**) and by Fisher’s exact test (**C**) and respective *p* values are indicated in each figure. Data on dietary habits are obtained by validated semiquantitative FFQs. * Expressed as the average number of returned capsules at the end of the study.

**Table 1 healthcare-10-02434-t001:** Comparison of primary and secondary outcomes (placebo vs. treated) at the initial visit and 3-month follow-up.

	Placebo n = 48	Treated n = 41	*p*	Placebo n = 48	Treated n = 41	*p*
	Initial Visit			3-Month Follow-Up Visit		
^1^ LSIL number of patients	48	41	1.000	44	2	<0.0001
^1^ HPV number of patients	18 (37.5%)	19 (46.3%)	0.5178	18 (37.5%)	19 (46.3%)	0.5178
^2^ hsCRP (mg/L)	0.86 (0.47–1.32)	1.77 (0.82–2.44)	0.0344	1.11 (0.61–1.98)	0.89 (0.39–1.68)	0.2408
^2^ FI (g/L)	3.30 (2.4–3.9)	3.9 (3.1–4.4)	0.0130	3.7 (2.7–4.5)	3.3 (2.6–3.8)	0.0437
^2^ SE (mm/h)	16 (12–23)	20 (15–26)	0.0785	20 (16–26)	16 (12–22)	0.0046

Results are expressed as medians (interquartile range). ^1^ Tested by Fisher’s exact test, comparing the placebo and the treated group of patients at the initial visit/follow up visit. ^2^ Tested by Mann–Whitney’s test, comparing the placebo and the treated group of patients at the initial visit/follow up visit.

**Table 2 healthcare-10-02434-t002:** Changes in primary and secondary outcomes between the initial and the 3-month appointment in placebo and treated group.

	Placebo			Treated		
	Initial	3-Month Follow-Up	*p*	Initial	3-Month Follow-Up	*p*
^1^ LSIL number of patients	48	44	0.1171	41	2	<0.001
^1^ HPV number of patients	18	18	1.000	19	19	1.000
^2^ hsCRP (mg/L)	0.86 (0.47–1.32)	1.11 (0.61–1.98)	<0.001	1.77 (0.82–2.44)	0.89 (0.39–1.68)	<0.001
^2^ FI (g/L)	3.30 (2.4–3.9)	3.7 (2.7–4.5)	<0.001	3.9 (3.1–4.4)	3.3 (2.6–3.8)	<0.001
^2^ SE (mm/h)	16 (12–23)	20 (16–26)	<0.001	20 (15–26)	16 (12–22)	<0.001

Results are expressed as medians (interquartile range). ^1^ Tested by Fisher’s exact test, comparing the values in the initial and the 3-month follow-up visit in placebo treated group. ^2^ Tested by Mann–Whitney’s test, comparing the values in the initial and the 3-month follow-up visit in placebo treated group.

**Table 3 healthcare-10-02434-t003:** Efficiency of 3-month ALA supplementation on the recovery of LSIL-positive patients (odds ratio and 95% Cl intervals).

	Placebo n = 48 (Final Visit)	Treated n = 41 (Final Visit)	Odds Ratio	95% Cl	*p*
Recovery (n)	4	39	0.004662	0.0008091–0.02686	<0.0001
LSIL (n)	44	2			

Results are expressed as number of participants (n). Tested by Fisher’s exact test.

## Data Availability

The data that support the findings of this study are available from the corresponding author [D.V.C.] upon reasonable request.

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
