# Peer review of "Effect of Alpha-Lipoic Acid Supplementation on Low-Grade Squamous Intraepithelial Lesions—Double-Blind, Randomized, Placebo-Controlled Trial"

_healthcare, 2022, doi:10.3390/healthcare10122434_

Round 1

Reviewer 1 Report

This study investigated the impact of supplementation with alpha lipoic acid (ALA), a potent antioxidant, in patients. All of whom had a diagnosis of low-grade squamous intraepithelial lesions; and where the effects of supplementation were observed after three months in the progression or regression of the disease. The authors found that ALA supplementation significantly reduced the proportion of patients with low-grade cytological abnormalities compared to placebo. In my opinion, the research is relevant, because in some way it was quite good that the vast majority of the members of the initial sample were able to retain it, however, perhaps from the planning of the project they were able to contemplate that the study times were extended until the 6 months; because the authors discuss this as one of its greatest limitations. I do not mean to say that their discussion is bad, on the contrary, but they contrast a lot with what has been done in other intervention studies regarding clinical trials investigating the efficiency of nutrition in inducing LSIL regression. One last observation would be that perhaps they could present the results in a more graphic way (schemes or figures if possible), trying to make everything much clearer and more attractive to readers.

Author Response

The authors would like to thank the reviewer 1 for his/her valuable insights regarding the structure and quality of the manuscript. Regarding the comments of the reviewer 1:

  1. Authors agree with you observation. Due to numerous reasons (primarily financial but also organizational, it was impossible for us to conduct a long-term (6-month) supplementation intervention study. We are aware that this presents the limitation of our study (as we stated in the discussion).

On the other hand, even though it is true that investigations of the impacts of food intake/diet are usually longer; intervention studies on clinical applications of dietary supplements (such as this) often investigate the efficiency of 3-month supplementation protocols. It is not usually the case with LSIL (or CIN1/CIN2), but we concluded that it was eligible to try the efficiency of the short-term supplementation. In the end, this fact makes our data standing out of other investigation, because it is one of the rare studies proving the efficiency of the short-term supplementation in LSIL.  It is our plan to build up on this research in terms of increased duration and wider spectrum of diagnostic parameters and biomarkers in the future research. In order to clarify, the sentence was added into discussion explaining that long term supplementation study was impossible to conduct due to financial and organizational limitations.

2.The authors agree with the reviewer’s observation on the visual quality of data presentation. Accordingly, adequate changes have been made in the manuscript.  In the original version of the manuscript the part of experimental design was already presented graphically, as the patient flow diagram in order to achieve adequate clearance. In the revised version of the manuscript, we additionally substituted Table 1 with Figure 1 as suggested by reviewer.  

Reviewer 2 Report

This is an interesting study, with promising clinical application.

Minor revision:

1. The notes of the tables are not clearly indicated.

2. Please provide raw data as well, and it is important.

3. If legal, please provide representative pictures of the physical and cellular tests.

Author Response

The authors would like to thank the reviewer 2 for his/her valuable insights regarding the structure and quality of the manuscript. Regarding the comments of the reviewer 2:

  1. The notes of the tables are not clearly indicated.

As suggested by reviewer 1, Table 1 has been substituted with Figure 1. The remaining tables (Table 1 and Table 2) have been changed to improve the clarity of presented data, as suggested by reviewer 2.

  1. Please provide raw data as well, and it is important.

Authors agree with the reviewer that providing raw data is important. Therefore, raw data were inserted in the revised version of the manuscript as Supplementary files (Table S1)

  1. If legal, please provide representative pictures of the physical and cellular tests.

Even though authors agree that providing representative photographs of physical and cellular tests would contribute to the quality of the manuscript, the Ethics Committee of the University of Zagreb, Faculty of Pharmacy and Biochemistry and the Ethics Committee of the University Clinical Centre Tuzla agreed that it would be unadvisable due to ethical concerns, particularly since it would not contribute to the scientific soundness of the manuscript. Namely, conducted tests are routinely applied and discussed in everyday clinical practice and as such their representative pictures are not relevant for proving the hypothesis of the work or discussing obtained data. Therefore, they were not included in the revised version of the manuscript.
